# Promoting teaching innovation of Chinese public-school teachers by team temporal leadership: The mediation of job autonomy and the moderation of work stress

Kai Li[1], Guiqin Zhu[2]*

**1** School of Management, Xi'an University of Architecture and Technology, Xi'an, Shaanxi, China, **2** School of Education, Chongqing Normal University, Chongqing, China

* qin7.7@126.com

## Abstract

This study examines the impact of team temporal leadership, leaders' behaviors regarding scheduling, allocating time resources, and coordinating team members, on teachers' innovative behavior. Questionnaire surveys on 2021 Chinese elementary and secondary public-school teachers show that team temporal leadership exerts a significant positive direct effect on teaching innovation and the effect can be facilitated through the mediation of job autonomy. Moreover, both the direct effect and the second-leg of the mediation effect are moderated by work stress. These suggest that, at least in certain educational settings, teaching innovation can benefit from leaders' appropriate scheduling and synchronization of time resources. The results also emphasizing the roles of job autonomy and work stress during this time-based team management.

**Data Availability Statement:** Data for this article can be accessed via https://osf.io/89xnh/. and https://github.com/gqzhucqu/leadershipplosone.

## 1. Introduction

Innovation, the process "leading to creative learning by generating and implementing new ideas, methods, tools, and content" [1, page 300], is at the heart of contemporary education. It promotes teacher effectiveness and stimulates children's motivation, learning, and curiosity [2]. As teachers are at the forefront of developing and implementing new ideas, their innovative behavior is critical for the effectiveness and sustainability of school organizations [3]. While innovative behavior can be self-initiated at the individual level, innovation in educational organizations is largely a collective, complex knowledge-based task in the orchestration between the teachers and leaders [4, 5]. Existing models and empirical studies have established close association between leadership and innovative work performance [6] and innovation in teaching in particular is contingent on leadership behaviors [3, 7, 8].

Innovation requires time. However, teachers nowadays face stressors on many fronts: increasing family responsibilities, task overload, student heterogeneity, evolving pedagogy and curriculum reforms, information explosion, just to name a few [4, 9]. Additionally, when teachers collaborate to improve instructional ideas and technologies, there arises the problem

**Funding:** Guiqin Zhu was supported by a grant from the 'National Planning Office of Philosophy and Social Science, China' (grant number: BHA180139). The funders had no role in study design, data collection and analysis, decision to publish, or preparation of the manuscript.

**Competing interests:** No conflicts of interest are declared. All authors have approved the final article.

of temporal diversity [10] due to individual differences in ability, time urgency, time-based orientations, and pacing. Time scarcity and temporal diversity pose critical challenges to teachers' innovative behaviors. Therefore, appropriate leadership from school supervisors are required to help teachers restructure time resources, synchronize rhythms, and avoid time-based conflicts.

In the last 20 years, the concept of temporal leadership [10] has received increasing attention from scholars and practitioners amid intense calls for integrating time-related issues into models of leadership [11, 12]. Temporal leadership refers to a unique module of leadership behaviors related to scheduling, allocating time resources, and coordinating team members in work rhythm, and helps the team achieve its goals by optimizing time resources [10]. This concept is considered a positive construct for organizational and employee performance. Importantly, in corporate and business settings, team supervisors' temporal leadership is positively associated with employees' innovative behavior [13, 14] and creativity [15]. It promotes innovation readiness [16], facilitates innovation, corporate venturing, and strategic renewal activities [17]. However, to the best of our knowledge, no report has yet been published on the impact of temporal leadership on teaching innovation.

Even if it can be observed that temporal leadership promotes teachers' innovative behaviors, other important questions arise such as: How is this realized through organizational processes and what are the boundary conditions? Drawing on existing evidence, particularly the work demands and resources theory [18, 19], we propose that temporal leadership positively affects teachers' innovative work behaviors and that its effect can be mediated by work resources such as work autonomy and moderated by work demands, including work stress. Work autonomy reflects the amount of freedom people have to decide what to do in their jobs and how to do it [19], and it is a precondition for innovative behavior in the literature [20]. Stress is pervasive in the workplace and fluctuates with time and circumstances. It significantly affects job performance and innovative behavior [21].

We tested this hypothesis by questionnaire surveys on a representative sample of elementary and secondary Chinese public-school teachers (n = 2021) and analyzing the data according to the well-established conditional process model [22]. The questions of interests are that: 1) if temporal leadership of principals positively predictive of innovative teacher behaviors; 2) if the effect of temporal leadership on teaching innovation could be mediated by job autonomy; and 3) whether the effect of temporal leadership on teaching innovation is moderated by work stress.

## 2 Background and hypothesis

### 2.1 The job demands-resources theory

As one of the most widely cited theories in the field of organizational behavior, the Job demands and resources (JD-R) theory [23, 24] identifies two main categories of work environment that influence work engagement, performance, and well-being: Job demands and job resources. The former refers to "those physical, social, or organizational aspects of work that require sustained physical or mental effort and are therefore associated with certain physiological or psychological costs" [23, page 9]. The latter refers to those physical, mental, social, or organizational aspects of work that are functional in (i) achieving work goals, (ii) reducing work demands and associated costs, and (iii) promoting personal growth, learning, and development" [23, page 9]. Typical examples of job demands are work pressure, emotional demands, and changing tasks, and typical examples of job resources are organizational support, job autonomy, and time control, allowance for failure [25, 26]. Job demands trigger a process of health impairment that impacts costs and leads to exhaustion and negative health

problems, whereas job resources trigger a motivational process that is positive for work engagement and performance [23]. Whereas these two sets of factors set different processes in motion, they also work together to influence organizational outcomes: Work demands can moderate the impact of work resources on motivation and engagement. Among a number of factors, autonomy at work and work stress are two important variables in the JD-R theory. Work stress is also considered as the most important work demand and work autonomy as the most important work resource in the influential demand-control model [19].

The JD-R theory has been well supported by empirical studies and modeling (see comprehensive reviews: [23, 24]. Levels of work resources have been found to be related to work engagement, intention to leave work [27] and project success [28]. High job demands predict burnout, health problems, and low well-being [29]. There is also empirical evidence of the interaction between job demands and job resources [23, 24]. The JD-R theory has been widely applied in recent decades in practices such as the JD-R monitor, organizational assessment, work design interventions, and personal resource interventions [24]. Critically, JD-R theory is also validated in innovative behaviors in organizations, workplaces, and educational institutions [30].

## 2.2 The team temporal leadership of school principals may affect teacher innovative work behavior

**2.2.1 Team temporal leadership.** Team temporal leadership (TTL) has been defined as "leader behaviors that aid in structuring, coordinating and managing the pacing of task accomplishment in a team" [10, p.492]. It concerns a unique dimension of leadership behavior that addresses time management in team activities [10, 31], such as scheduling, time synchronization, and time resource allocation. As a multidimensional construct, temporal leadership includes two core components: task-oriented and relationship-oriented temporal leadership [31]. The task-oriented dimension refers to leaders' behaviors that help structure, coordinate, and manage the pace of task completion, including reminding members of deadlines, scheduling time for contingencies and problems, and synchronizing members to get work done on time [10, 31]. The relational dimension describes the behaviors of leaders that support members in managing time in the team, such as consulting with team members before making a decision about the schedule, dealing with time desynchronization among members, and praising members' performance on schedule [31]. Previous studies have found it cushions the blow of temporal diversity on team performance [10, 32], and reduces time-based conflict when people with different styles of pace or time perspective work together [32]. It promotes employee followership [33], competence [33], and professional engagement [34].

**2.2.2 Teaching Innovation.** Some researchers conceptualize teaching innovation as "teachers' receptivity, openness, and willingness to embrace change" [4, page 1], and regard it as a critical factor in teachers' innovative behavior and a prerequisite for innovation and change. Blândul [35] identifies three main indicators of teaching innovation: information content, teaching materials, and teaching/assessment methods. In recent studies from East Asia, Wei et al. [36] propose a five-factor model of "creative teaching", including "interactive discussions," "open mind," "problem-solving," and "multiple-level teaching" and "independent learning". Zhu et al. propose four core competencies, including "learning competence, pedagogical competence, social competence, and technological competence" [37, page 10], and summarize three typical trends for innovative teaching and learning: "integrated use of information and communication technologies, the adoption of student-centered learning, and the use of collaborative learning approaches" [38, page 2]. Hung and Li advocate that "the content of innovative teaching can be categorized into five dimensions, namely idea thinking, curriculum content,

teaching materials, teaching methods, and multifaceted assessment." [39, page 1037]. This is in line with Hou's definition of teaching innovation, which proposes five similar dimensions of teaching innovation: "thinking innovation, content innovation, method innovation, resource innovation, and assessment innovation" [40, page 30].

Although these definitions vary to some extent, they agree on some dimensions that are generally accepted. Teaching innovation is generally understood to be the innovative behaviors of teachers in their workplace, encompassing key components such as generating, promoting, implementing creative ideas, materials and tools for instruction and evaluation. The purposes of these activities include promoting teaching effectiveness and fostering students' development. These various proposals suggest that teaching innovation is more than just incorporating new technologies into the classroom and promoting achievement; it also involves a new ideology that educates students to become independent and responsible learners and social agents. It is also worth noting that the studies agree that teachers' activities are embedded in interactive social and pedagogical environments. One's own teaching innovation should be intertwined with that of other members, although it could also be initiated at the individual level [4].

Innovation in teaching is associated with endogenous variables such as attitudes [41] and motivation [42]. It is also enhanced or affected by innovation atmosphere [40], work autonomy [4], organizational support [9], and most importantly, the principal's leadership style [43]. Transformational [40, 44] and transactional leadership [40] have been found to be positively associated with teachers' performance in developing and implementing creative ideas in schools. Decentralized leadership by school leaders encourages teachers to adopt network-based innovations in the workplace [45]. Empowering leadership from school leaders [46] encourages teachers to explore, which in turn facilitates their innovative work behaviors. Visionary leadership [47], and servant leadership [48] are also considered conducive to creative teaching behaviors.

**2.2.3 The impact of temporal leadership on teaching innovation.** Although there are no publications on the role of temporal leadership in innovation theory, existing studies in business and industry have shown that temporal leadership of team leaders is positively associated with innovative behavior [14, 49, 50] and creativity [15], and facilitates business innovation, venturing, and strategic renewal activities [17]. From a team process perspective, temporal leadership positively impacts team transition, action, and interpersonal processes. Team transition describes the team's behaviors related to mission analysis, goal setting, and strategy formulation and planning. Team action assesses team members' engagement in accomplishing tasks, such as monitoring progress toward goals, backup behaviors, and coordination. Interpersonal processes assess the degree of trust and support team members share in resolving task and emotional conflicts [51]. When team leaders exhibit strong temporal leadership behaviors, they reduce time-related problems such as procrastination, along with decrease burnout and conflict [16], and ultimately increase diversity benefits in terms of time-based urgency and pace style [52]. Leaders who provide temporal coordination or support individuals in addressing time-related issues encourage teams to develop an integrated approach to time, thereby increasing task clarity [53] and synchronizing organizational dynamics, which better positions the team for higher quality productivity [54].

With this in mind, it is plausible that temporal leadership can avoid team conflict and increase the temporal smoothness of the team, which would help teachers adjust their pace to external deadlines and increase efficiency. Temporal synergy in the team could also minimize problems such as time-based fragmentation, time-based uncertainty, and time-based ambiguity, giving teachers more time-based resources and autonomy for innovative activities. We

therefore propose that principals' temporal leadership, a type of organizational control, may act as a positive labor resource for teaching innovation:

H1: *Temporal leadership is positively predictive of teaching innovation.*

## 2.3 The mediation of job autonomy

**2.3.1 Job autonomy.** Previous studies on innovation in business and the economy show that the influence of temporal leadership is mediated by factors such as temporal conflicts [16], pro-social rule violation for efficiency, and vitality [50]. Similarly, the effects on teaching innovation are also mediated by external factors. In this study, we choose to focus on workplace autonomy, one of the key workplace resources in influential theories of organization and work environment [19]. Autonomy in the workplace is also referred to as job control in other influential theories, such as demand control theory [19, 55] and the job characteristics model [56, 57]. The concept of "autonomy" can be interpreted at the individual, collective, or state level [4]. At the individual level, which is the focus of this study, workplace autonomy reflects the amount of freedom a worker has in deciding what to do at work and how to do it [19]. Other researchers propose a four-dimensional model: autonomy of work method, of work scheduling, of work time, and of work location [58]. Autonomy is one of the central motivating factors for organizational behavior and leadership in the work characteristics model [56, 57] and the self-determination theory [59, 60].

**2.3.2 Job autonomy as an antecedent of innovative behavior.** Autonomy is a critical for employee engagement and innovative behavior [4, 61]. It is a precondition for innovation [62], and can have a direct effect on innovative work behaviors; however, it can also have an indirect effect by mediating job engagement, whereby job autonomy increases job engagement, which in turn promotes innovative work behaviors, such as idea generation and implementation [63]. Autonomy is thought to influence innovation not only through the degree of autonomy granted by organizations, but also through the autonomy individuals gain for themselves [20]. As a multidimensional construct, all of its dimensions are positively related to innovative behavior, with the role of autonomy in work methods and work locations being particularly significant [58]. Giving employees the freedom to determine how tasks should be performed can increase their willingness to develop and implement new ideas and technologies [62].

In terms of teaching, autonomy also has a number of positive functions. It is a catalyst for teachers' professional competence, a source of effectiveness, and a source of creativity, experimentation, and diversity [4]. Teachers' weekly work autonomy, along with other work resources, positively correlates with weekly engagement, which in turn increases their work performance [64]. Job autonomy also influences teachers' collective innovativeness by mediating their participation in professional learning activities [4]. In contrast, a lack of autonomy discourages teachers from engaging in innovative instructional activities, such as implementing new pedagogical approaches to teaching English [65].

**2.3.3 The myth between temporal leadership and job autonomy.** Autonomy in the workplace can be promoted by leadership styles [13, 66]. For instance, ethical leaders provide employees with greater autonomy at work by providing social cues embedded in values, work requirements, and expectations [13]. Some studies [67] found that job autonomy fully mediates the effect of transformational leadership on work engagement. In contrast, workplace autonomy is impaired in leadership styles such as lean production [68], which control all decisions and require little input from employees. In educational systems, teachers' workplace autonomy can be shaped by educational reforms and constrained by policies such as "Schemes

of Work" set by local and national boards, the state, and school-based oversight and monitoring mechanisms [69].

While the pathway from workplace autonomy to teaching innovation is well documented, the impact of temporal leadership on staff workplace autonomy remains unresolved. Job autonomy in organizational management requires freedom from external control, temporal leadership appears to affect job autonomy, however, it does not mean that one's work behavior can be completely isolated or independent of other team members. Unsynchronized time structures can lead to fragmented work and higher burnout. Indeed, workplace autonomy and team time management are not incompatible in modern knowledge work. One can enjoy strong autonomy while being bound to the macro team framework [70]. Time is an important dimension of work autonomy. Effective time management by managers can avoid planning uncertainties and increase task clarity. Team members can then enjoy a high level of time commitment and gain more free time for innovative activities [58, 71]. Therefore, we posit the following hypothesis:

H2: *Team temporal leadership promotes teachers' job autonomy.*

H3: *Job autonomy mediates the impact of temporal leadership on teaching innovation.*

## 2.4 The moderation of work stress

Because stress is ubiquitous in the workplace, it is a classic and fundamental work demand in models of organizational behavior. In the theory of job demands and resources [19], job demand is measured by job performance pressures, psychological stressors associated with workload, unexpected tasks, work-related personal conflicts, unemployment or job anxiety, career development, and physical stressors. Stress reflects "individuals' perceived demands versus resources" [72]. The American Institute of Stress defines stress as "a state or feeling experienced when a person perceives that demands exceed the personal and social resources that the individual can mobilize" and considers it an intervening variable with antecedent causes and behavioral consequences [73].

According to the theory of work demands and resources [24], work demands and work resources interact to influence health, well-being, and work engagement. The hypothesis is that work resources are most useful for maintaining work engagement under conditions of high work demands (e.g., workload and unfavorable physical environment) [23]. Accordingly, the innovation-enhancing effect of temporal leadership is stronger when time pressure is high [49] or when the employee has a higher synchrony preference [14]. The effect of autonomy on innovation also depends on boundary conditions. It is moderated by working conditions, employees' prior experience or skills, and self-control [61]. However, little is known about how the effects of temporal leadership and work autonomy on teaching innovation might be moderated by boundary conditions.

The literature has documented complex functions of work stress on organizational performance and innovation. Behavioral and psychological models have identified two typical copying behaviors under stress: people revert to rigid and routine responses from previous experiences [74] or make risky and erroneous decisions [72]. Stress is perceived by some researchers as an innovation killer [75, 76]. Under stress or when resources are scarce, people shift their attention to the problem at hand [77], reinforcing the urgency and importance of what is at hand at the expense of the future [74, 78, 79]. Stressful events may prompt reflexive or habitual reaction [80], and impair the flexible adaptation to changing environment [76]. At the group level, some stressors such as time scarcity reduce team trust building [81], impair

team knowledge transfer effectiveness [81]. However, there are also studies establishing positive effect of workspace stress on innovative work behavior [21]. Individuals under stress may be compelled to seek solutions and overcome challenges in an active problem-solving mode to self-develop and transform [82]. Under circumstances characterized by high job demands, stress could lead to higher levels of motivation, learning, and innovative behavior [82–84]. We hence predict:

H4: *Work stress moderates the effect of temporal leadership on teaching innovation.*

## 3. Methods

### 3.1 Participants

A total of 2021 valid samples were recruited from teachers in over 355 elementary or secondary public schools in 19 provinces in mainland China using a cluster sampling method. The response rate was 80.8%. This included 436 men and 1,585 women with a mean age of 36.67 years (*SD* = 8.92). 1% of them were with a diploma of high school or below, 27.1% with associate degree, 71.4% were with bachelor degree, and the other 0.5% were with master degree or higher. In term of work experience, 5.9% of them have taught for less than 1 year (1 inclusive), 18.3% for 1 to 5 years (1 inclusive), 19.8% for 5 to 10 years (10 inclusive), 25.3% for 10 to 20 years (20 inclusive), 24.3% for 20 to 30 years (30 inclusive), 6.1% for 30 to 40 years (40 inclusive). The number of participant teachers in these schools ranged from 1 to 53. In distributing the questionnaires, we treated the school unit as one group. First, the principal asked teachers to gather at a predetermined location. A trained research assistant then formulated the detailed guidelines, explained the implications of the survey, and ensured that anonymity was maintained. Teachers were asked to respond to all questions in the questionnaire according to their natural feelings. Important socio-demographic variables such as gender, age, and education level were also asked. The survey was conducted in anonymity, the participants were encouraged to respond to the questions but allowed to withdraw or skip the items they were reluctant to answer. The study was approved by the Ethic Committee of Xinyang Normal University (No. XYEC-2021-009), all the teachers gave written consents for participation.

### 3.2 Measurement and construct validation

**3.2.1 Team Temporal leadership.** The teachers were asked to rate their supervisor's temporal leadership, such as commitment to scheduling, synchronizing, and allocating temporal resources, using the scale developed by Myer and Mohammed [85]. The scale contained 10 items, with five items for each of the two dimensions: task-oriented temporal leadership and relationship-oriented temporal leadership. A sample item for the task-oriented dimension was "to what extent does your team leader push members to complete subtasks on time?" and a sample item for the relationship-oriented dimension was "to what extent does your team leader praise (commend) team members for completing work on time?" Each item was scored on a 5-point scale, ranging from 1 ("not at all") to 5 ("a great deal"), with a higher score indicating a higher level of team temporal leadership. The Cronbach's α for this scale was 0.929 for the current sample.

**3.2.2 Teaching innovation.** We measured teaching innovation using Chen's questionnaire [86] for primary and secondary teachers as a reference. There were five dimensions, including innovation of ideas and thinking, of teaching content, of teaching methods, of teaching materials, and of multiple assessment. Hou [87] later adapted and revised the 25-item questionnaire by adding the specific situation of teachers. As the innovation in teaching

community is fast evolving and some strategies listed in Hou's questionnaire [87] were not innovative any more, we consulted three experts in the education and management fields and deleted three items that did not fit for our target participants based on their consensus. We then tested the reliability and validity of the questionnaire for use in the context of rural education. The final version of the questionnaire comprised 22 items scored on a 5-point scale, with higher scores indicating higher teaching innovation. Confirmatory factorial analysis of all 2021 participants repeated the five similar dimensions of teaching innovation defined by Hou's [40], including "thinking innovation, content innovation, method innovation, resource innovation, and assessment innovation". It also revealed a satisfactory fit to the five-factor model, chi-square = 2340.663, $df$ = 199, p < 0.001, NFI = 0.926, RFI = 0.906, IFI = 0.932, TLI = 0.913, CFI = 0.932, RMSEA = 0.073. For the current sample, the Cronbach's α was 0.956.

**3.2.3 Job autonomy.**   The 7-item scale developed by Kirmeyer and Shirom [88] was used to assess perceived work autonomy. Those items rated on a 5-point scale, with higher scores indicating higher levels of autonomy at work. An example item was "I can decide how to perform my job based on my work style." For the current sample, the Cronbach's α was 0.868.

**3.2.4 Work stress.**   Work stress was measured by the 11-item scale developed by Cavanaugh et al [89], which contained two dimensions: challenge-related stress and obstacle-related stress. The teachers were asked to rate how stressed they felt in situations described by each item, for example: "the number of projects or tasks I have". Cronbach's α was 0.929 for the present sample.

## 3.3 Data processing

Statistical analyzes were performed using IBM SPSS 20.0 software in the following steps: (step 1) descriptive statistics and correlation analysis for each variable; (step 2) common method bias analysis; (3) regression analyses (the model 4) to assess the mediation of job autonomy on the effect of temporal leadership on teaching innovation; and (4) a conditional process analysis (model 15) to examine the moderation of job stress on the mediated effect of temporal leadership. To test the possibility of questionnaire bias (common method bias, CMB) [90], we anonymized all respondents and included all items of the above-mentioned 4 questionnaires and performed a principal component analysis. In analysis 3 and 4, the number of the Bootstrap samples were set to 5000 with 'Bias Corrected'. The confidence level for confidence intervals was 95% as default. In previous studies, age and gender exhibit nonsignificant relationships with the two dimensions of team temporal leadership [10, 50, 85], but there are gender differences in job autonomy [91] and teacher innovation [4], as well as age difference in job autonomy [4] and innovation [4]. So, it is important to explore the relationship between our core variables of interests with gender and age under controlled. We therefore included these two important demographic variables as covariates in analyses 3 and 4 followed previous studies [4, 13].

## 4. Results

### 4.1 Mean values, standard deviations, and correlation matrixes for each variable

Table 1 presents the means, standard deviations, and correlation matrices for each variable. We found that gender, age, and educational background were significantly correlated with some variables. Therefore, we controlled for each of these variables in a subsequent regression analysis, finding significant correlations with temporal leadership in all cases. Specifically,

Table 1. Mean values, standard deviations, and correlation matrixes among variables.

| | $M \pm SD$ | 1 | 2 | 3 | 4 | 5 |
|---|---|---|---|---|---|---|
| 1. Gender | 0.79±0.411 | — | | | | |
| 2. Age | 36.77±8.714 | -0.331*** | — | | | |
| 3. Temporal leadership | 3.404±0.813 | 0.072** | -0.097*** | — | | |
| 4. Job autonomy | 3.359±0.719 | 0.017 | -0.025 | 0.403*** | — | |
| 5. Work stress | 3.152±0.796 | -0.122*** | 0.035 | -0.170*** | -0.227*** | — |
| 6. Teaching innovation | 4.171±0.466 | 0.098*** | 0.031 | 0.303*** | 0.244*** | -0.069** |

Note:

* $p < 0.05$,

** $p < 0.01$,

*** $p < 0.001$.

Gender was the dummy variable. Female = 0, Male = 1. The mean value represents the proportion of females.

temporal leadership was significantly and positively correlated with teaching innovation and work autonomy, but negatively correlated with work stress. Additionally, there was a critical correlation between teaching innovation and work stress. Finally, there was a significant and negative correlation between work autonomy and work stress.

## 4.2 Common method bias

Common method bias analysis revealed two factors with eigenvalues greater than one. The first factor extracted explained 27.57% of the total variation before rotation and 21.6% after rotation, which was far below the critical value of 40% and suggested no obvious CMB.

## 4.3 Test of the conditional process model with work stress as a moderator

All variables were standardized to avoid multicollinearity. We then tested Hypothesis 1 through a linear multiple regression analysis with teaching innovation as the dependent variable and team temporal leadership over time as the independent variable, while the control covariates included age and gender. The results showed a significant positive prediction of temporal leadership on teaching innovation ($\beta = 0.305$, $t = 14.111$, $p < 0.001$). We then constructed a mediation model (Model 4) using ordinary least squares path analysis in PROCESS 3.0 [92] to test Hypotheses 2 and 3. The significance of the indirect effect was estimated using 5000-percentile bootstrap samples, and the mediation effect size was indexed by the partially standardized indirect effect (Fig 1). As predicted, temporal leadership was positively predictive of job autonomy ($\beta_a = 0.404$, $t = 19.683$, $p < 0.001$, [95% bootstrapped CI: 0.321 to 0.392]) and job autonomy was positively associated with teaching innovation ($\beta_b = 0.148$, $t = 6.482$, $p < 0.001$, [CI: 0.677 to 0.126]). This resulted in a partial but significant mediation effect of job autonomy ($ab = 0.060$, $p < 0.05$, [CI: 0.040 to 0.080]), leaving the direct effect of temporal leadership on teaching innovation $c' = 0.245$ ($t = 10.726$, $p < 0.001$).

We then tested the moderated mediation effect of work stress (Fig 2). Model 15 was chosen to test the hypothesis that work stress influences the direct effect of temporal leadership on teaching innovation as well as the second part of the mediation pathway. Regression analysis revealed the positive relationship between job autonomy and teaching innovation ($\beta = 0.399$, $t = 5.055$, $p < 0.001$, [CI: 0.244 to 0.553]). The main effect of work stress on teaching innovation was not significant ($\beta = 0.039$, $t = 1.441$, $p > 0.092$ [CI: -0.014 to 0.092]). However, the magnitude of the interaction between work stress and work autonomy was significantly negatively associated with teaching innovation ($\beta = -0.076$, $t = -3.224$, $p = 0.001$, [CI: -0.122 to

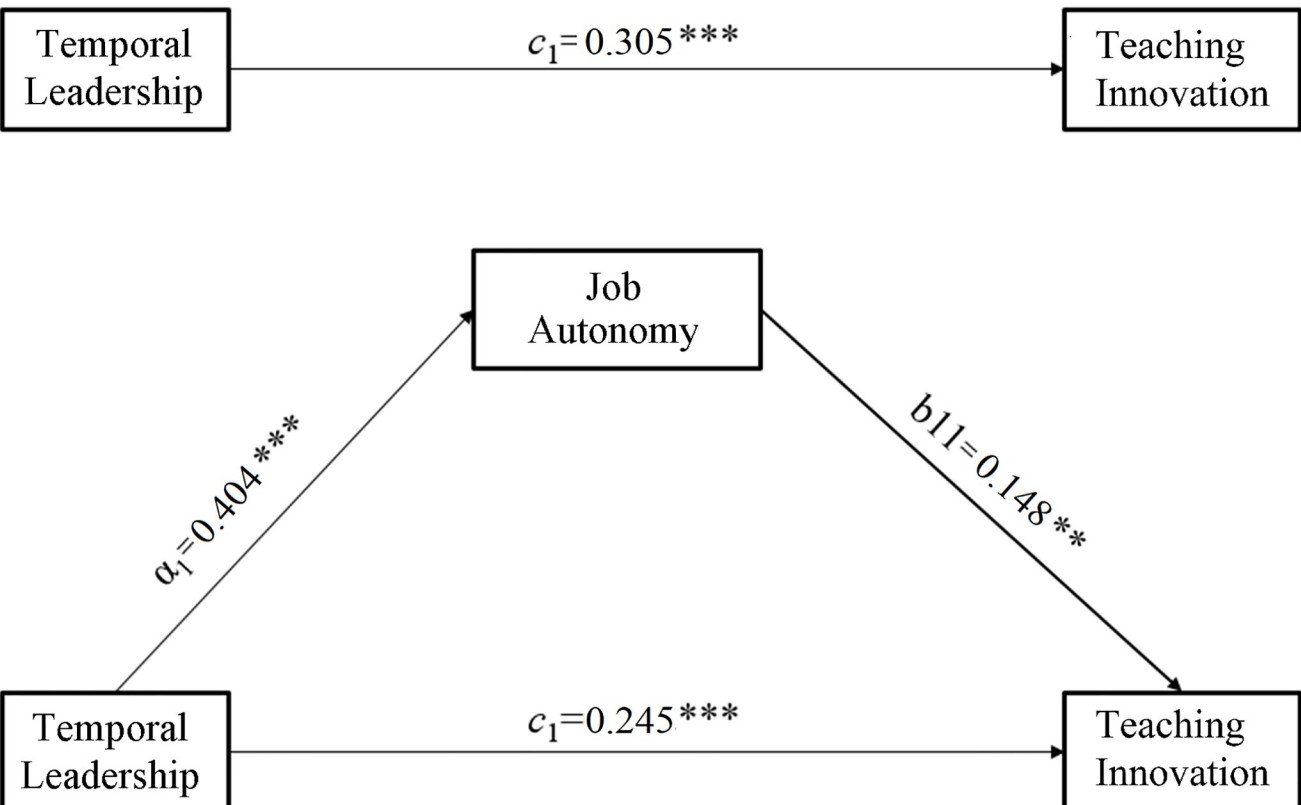

**Fig 1. Job autonomy mediates the relationship between temporal leadership and reaching innovation.** ** $p<0.01$, *** $p<0.001$.

-0.030]). These results support Hypothesis 4, that work stress moderates the second limb of the mediation effect of occupational autonomy. Meanwhile, we observed a significant moderation effect of work stress on the direct effect of temporal leadership on teaching innovation, as their interaction was significantly and negatively associated with teaching innovation ($\beta$ = -0.079, $t$ = -3.094, $p$ = 0.002, [CI: -0.129 to -0.029]).

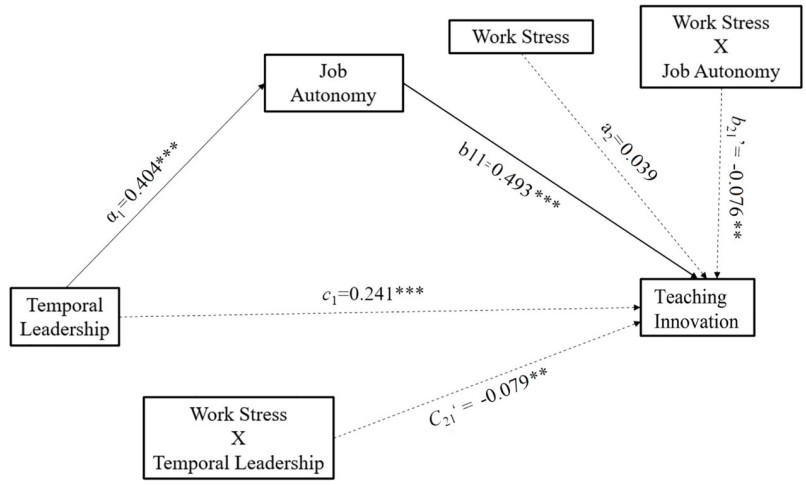

**Fig 2. The moderation of work stress.** ** $p<0.01$, *** $p<0.001$.

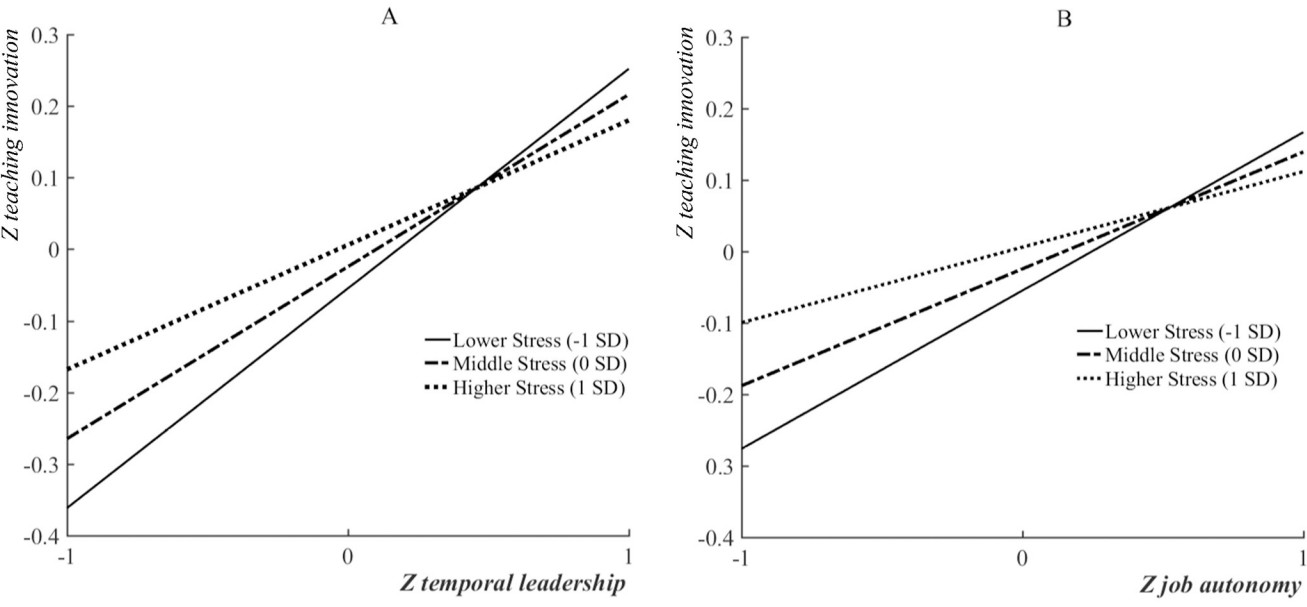

**Fig 3. Simple slope analysis on the moderation of work stress.**

We then constructed slope diagrams to visualize the moderation effect of work stress. Previous studies have reported complex profiles, either an inverted U-shape or a linear shape, of the effect of stress on work performance. Taking this into consideration, we categorized work stress into three levels: high, medium, and low-level groups, based on standard deviations -1, 0, and 1 from the mean. The influence of temporal leadership on teaching innovation varied among the different stress levels, as did the influence of job autonomy on teaching innovation. There are significant conditional effects of temporal leadership at all three stress levels (Fig 3): lower stress (*effect* = 0.308, t = 10.199, p < 0.001, [CI: 0.249, 0.367]), medium stress (*effect* = 0.245, t = 10.548, p < 0.001, [CI: 0.200, 0.290]), higher stress (*effect* = 0.182, t = 5.886, p < 0.001, [CI: 0.121, 0.243]). However, the significant interaction implies that the conditional effect of temporal leadership is largest in the lower stress situation, and smallest in the higher stress situation. This indicates that temporal leadership facilitates teaching innovation at all three stress levels; however, the facilitation effect decreases as work stress increases.

There are significant conditional effects of job autonomy in all three levels of stress: lower stress (*effect* = 0.089, [CI: 0.059, 0.120]), middle stress (*effect* = 0.065, [CI: 0.044, 0.089]), higher stress (*effect* = 0.040, [CI: 0.013, 0.068]), leading to a significant moderated mediation of these two factors, *effect* = -0.031, [CI: -0.057, -0.008]). The conditional effect of temporal leadership is largest in the lower stress situation and smallest in the higher stress situation. This implies that workplace autonomy promotes teaching innovation at all three stress levels; however, the promoting effect decreases as work stress increases.

## 5. Discussion

Although time has emerged as an important dimension in renovated leadership theories [93] and is perceived as a critical factor that continuously impacts innovation [94, 95], few empirical studies exist on the relationship between temporal leadership and teaching innovation. To the best of our knowledge, this study is the first to link the two important constructs through a questionnaire survey of a representative sample of primary and secondary teachers. The results show a significant positive direct effect of temporal leadership on teachers' innovative work

behavior in a large sample of Chinese elementary and secondary school teachers. In Chinese public-school settings, temporal leadership facilitates teaching innovation via the mediation of job autonomy; it also facilitates job autonomy, which, in turn, enables teaching innovation. In addition, both the direct and indirect effects are susceptible to work stress. These findings are an important contribution to educational management and temporal leadership literature and have important theoretical and practical implications.

## 5.1 Theoretical Implications

The first important finding is that, in Chinese elementary and secondary teachers, team temporal leadership has a significant positive impact on teaching innovation, at least in certain education systems. This is consistent with previous findings that team temporal leadership positively affects innovation and work performance [15, 17] in business and economics. Given that there has not been many research on team temporal leadership on education and teaching particularly, our work represents an important extension of research in temporal leadership. It suggests cross-domain consistency regarding the impact of temporal leadership on innovation behavior.

The importance of autonomy for creativity and innovation is well known. For instance, freedom from external pressures stimulates interest in the task itself, increases intrinsic motivation, and thus promotes innovative behavior [60]. However, previous research on autonomy promotion has focused primarily on strategies, such as empowering employees to make independent decisions [96], encouraging exploration and innovation [97], and effective leadership style [66, 67]. Little is known about the effect of team leaders' temporal leadership on employees' job autonomy despite that time is critical to the management of an organization and time-based autonomy is a core dimension of job autonomy. While replicating previous effects of job autonomy on teaching innovation [4, 65], our study shows, for the first time, that temporal leadership favors job autonomy. Therefore, this finding enriches the modulating factors of job autonomy.

Autonomy means the freedom to make decisions and implement ideas without being constrained by external pressures [19, 88]. Because team temporal leadership involves control behaviors from managers such as scheduling and time allocation, as well as dealing with individual differences in task pace, one might speculate that temporal leadership may impair autonomy and innovation. In contrast to this speculation, our results show a facilitative effect of temporal leadership, suggesting that control and autonomy are not mutually exclusive, but can coexist. Recently, the interesting topic of bounded autonomy in knowledge work has emerged, and studies are called for to explore the methods by which it can be established. Researchers argue that since the work process is embedded in multiple social and organizational relationships, a high degree of individual decision-making autonomy and a high degree of connectivity and interdependence must coexist [98]. Our study is a response to this call by suggesting that temporal leadership can be used to realize bounded autonomy in organizations.

Why could team temporal leadership render such a positive effect on innovation? According to the theoretical constructs of temporal leadership [52], effective temporal leadership involves planning and allocating time resources with full consideration of individual differences in time-based orientation, pace, rhythm, and task difficulty, as well as appreciating and encouraging team members as they grapple with time-related work problems. Temporal leadership is not synonymous with rigid time management that turns a blind eye to individual differences among team members. Here, we propose two possible ways in which effective temporal leadership promotes work autonomy: 1) temporal leadership provides employees

with freer time resources by minimizing time fragments and increasing task clarity; and 2) temporal leadership resolves team conflicts, thereby ensuring smooth teamwork and bound autonomy that is fair to both members and the organization. Future studies can explore this proposition to uncover the processes underlying the effects of temporal leadership on work autonomy.

The moderation of work stress describes a boundary condition for the effects of temporal leadership and job autonomy on teaching innovation. It also supports the theory of job demands and resources [23] and the theory of job demands and control [19] by including temporal leadership as an element of job demands. Our study shows that work stress moderates the direct impact of temporal leadership on teaching innovation, as well as the effect of job autonomy on teaching innovation. The results showed that the benefits of higher temporal leadership on teaching innovation were smaller at higher levels of stress, but much larger at lower levels of stress. This suggests that the benefits of temporal leadership for teaching innovation become smaller as work stress increases. Previous studies found that a higher effect of temporal leadership was mediated by the amount of temporal conflict in the group. Benefits were higher when group temporal conflict was low, but decreased as time-based conflict increased [32]. When time-based diversity is higher, higher temporal leadership leads to the highest time-based conflict, while higher temporal leadership leads to the lowest time-based conflict when time-based diversity is lower [32]. When team temporal leadership is strong, the indirect effect of perceived time pressure on team performance is usually positive, whereas when team temporal leadership is weak, the indirect effect is positive when perceived time pressure is low, and negative when perceived time pressure is moderate to high [51].

## 5.2 Practical implications

Our findings provide three practical suggestions for educational management and, more broadly, leadership. First, our study clearly shows that temporal leadership can be useful for school leaders. It can be used to promote job autonomy and support innovative work behaviors among their staff, such as teaching innovation. Leaders can use levers, such as setting deadlines for tasks and encouraging their staff to stick to the schedule, to avoid time pressure to innovate [94]. It is important that managers avoid rigid task control when performing these activities; rather they should adapt their management practices to individual differences in time use [15]. In other words, they should pay attention to individual differences in team members' time orientation, pace, skills, and time urgency. Team members' appreciation of their time-related activities is also important. These findings are also significant in selecting team members when assembling the group. Selecting faculty with shared temporal cognition would be helpful because there is evidence that shared temporal cognition can, to some extent, substitute for the effect of temporal leadership on team work performance [54, 99]. Second, when using team temporal leadership as a catalyst for teaching innovation, leaders should be aware of the moderation of stress. Our results show that the effect size of temporal leadership is largest at lower levels of work stress, but smallest at higher levels of work stress. This means that leaders can use temporal leadership to promote teaching innovation at any stress level, but they should adjust their expectations accordingly. Third, leaders are advised to pay attention to motivational stimuli when promoting innovative behavior through team time management. Our study shows that workplace autonomy is a positive predictor of teaching innovation and that it also partially mediates the effect of temporal leadership on teaching innovation. This implies that in order to promote teaching innovation, school leaders can mobilize factors that promote workplace autonomy and thus could magnify the outcome of their leadership behavior. It is expected that this would also be true for innovative activities in other areas.

## 5.3 Limitations and further directions

Despite the significance and strengths, the results of our study should be interpreted with caution. First, our study is based on observations of primary and secondary public-school teachers in mainland China. Chinese culture emphasizes social harmony, conformity, and collective thinking, which is quite different from the Western where individual contribution is more valued. There are also significant differences between countries in the systems of educational administration and pedagogy. For example, teachers in different countries vary in their understanding about teacher autonomy and pedagogical tasks [100]. It is also worth noting that public demonstrate aspects of uniqueness from other institute types or organizations, such as private or charter schools where, according to some theories, competition is more prevalent and innovative behaviors are more emphasized. Thus, the question remains as to whether the same relationship between temporal leadership and job autonomy, temporal leadership and innovative behavior can be observed in other cultures, educational systems, and social sectors. Additionally, our study was based on teachers' self-assessment about teaching innovation and their supervisors' team temporal leadership. Though subjective ratings could be positively associated with objective measurement [101], it is still important to explore if our findings could be replicated with actual team temporal leadership and objective indices of teaching innovation. For instance, indices such as school performance [102] and contextualized behavior-based leadership assessment rubric [101] can be referred for future studies. Finally, some studies have shown that innovation can be hindered when behavioral norms are strictly enforced or when lesson plans are far too detailed, for instance, micromanagement [20]. Thus, the question arises as to what constitutes effective temporal leadership and what the difference is between the quality and the intensity of temporal leadership. It is also interesting to explore in depth the roles of discrete aspects of job demands and job autonomy (such as allowance for failure in innovation) or teachers' self-efficacy. Future studies are needed to make these distinctions.

## Supporting information

**S1 Data.**
(SAV)

## Author Contributions

**Conceptualization:** Guiqin Zhu.

**Data curation:** Kai Li, Guiqin Zhu.

**Formal analysis:** Kai Li.

**Funding acquisition:** Guiqin Zhu.

**Investigation:** Kai Li.

**Methodology:** Kai Li.

**Writing – original draft:** Kai Li, Guiqin Zhu.

**Writing – review & editing:** Kai Li, Guiqin Zhu.

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
