## [Decision Letter · Decision Letter 0]

18 May 2022

PONE-D-22-11897Promoting Teaching Innovation by Team Temporal Leadership: The Mediation of Job Autonomy and the modulation of Work StressPLOS ONE

Dear Dr. Guiqin Zhu,

Thank you for submitting your manuscript to PLOS ONE. After careful consideration, we feel that it has merit but does not fully meet PLOS ONE’s publication criteria as it currently stands. Therefore, we invite you to submit a revised version of the manuscript that addresses the points raised during the review process.

We look forward to receiving your revised manuscript.

Kind regards,

Rogis Baker, Ph.D

Academic Editor

PLOS ONE

Journal Requirements:

When submitting your revision, we need you to address these additional requirements.1.

2. PLOS ONE does not copy edit accepted manuscripts (https://journals.plos.org/plosone/s/criteria-for-publication#loc-5). To that effect, please ensure that your submission is free of typos and grammatical errors.

4. Please remove your figures from within your manuscript file, leaving only the individual TIFF/EPS image files, uploaded separately.  These will be automatically included in the reviewers’ PDF

Reviewers' comments:

Reviewer's Responses to Questions

**Comments to the Author**

1. Is the manuscript technically sound, and do the data support the conclusions?

Reviewer #1: Yes

Reviewer #2: Partly

Reviewer #3: Yes

Reviewer #4: Yes

Reviewer #5: Yes

Reviewer #6: Yes

Reviewer #7: Yes

2. Has the statistical analysis been performed appropriately and rigorously? 

Reviewer #1: Yes

Reviewer #2: Yes

Reviewer #3: Yes

Reviewer #4: Yes

Reviewer #5: Yes

Reviewer #6: Yes

Reviewer #7: Yes

3. Have the authors made all data underlying the findings in their manuscript fully available?

Reviewer #1: Yes

Reviewer #2: Yes

Reviewer #3: Yes

Reviewer #4: No

Reviewer #5: Yes

Reviewer #6: Yes

Reviewer #7: Yes

4. Is the manuscript presented in an intelligible fashion and written in standard English?

Reviewer #1: Yes

Reviewer #2: Yes

Reviewer #3: No

Reviewer #4: Yes

Reviewer #5: Yes

Reviewer #6: Yes

Reviewer #7: Yes

5. Review Comments to the Author

Reviewer #1: This is quite a novel research area as there hasn't been many research on temporal leadership in education and teaching particularly. The literature review is quite comprehensive exploring important and key components of the research.

The findings of this research are an important contribution to literature on educational leadership and management, and temporal leadership and have important theoretical and practical implications. The application and pertinence of the paper are obvious, but the academic and theoretical value needs to be further improved through deeper research. Despite the comprehensiveness of the research, It may still be doubtful whether the findings of the study are generalizable or not.

Reviewer #2: I thank the authors for their efforts and will like to provide the following feedback for their consideration to enhance the paper:

Introduction

Some quotations need page number

Some references are old and need updating, e.g. 2003, 1979, 2011 and others

In the abstract, 2021 surveys administered but 2017 indicated in the introduction. This needs to be fixed

Authors already proposed and concluded in the initial phase of the introduction before spelling out what they need to test as hypotheses.

Theoretical background and hypotheses?

Quotations need page number in in-text citations and more recent publications will add value to the paper

Some hypotheses are tested and accepted. That is not enough to generalize. So, contextualizing this study can add value to the work

Hypotheses 1 - 4 … in the study context? There is no need for splitting H4 into two.

3 Methods

Participants – Methods used and justifications? Justification of the sampling technique? Other ethical issues considered?

Justifications for each approach described and data analysis strategies

Results should focus on the findings. Strategies for minimizing biases to be transferred to the methods section

No hypotheses have to do with demographics such as gender and age. Authors should decide on what they want to explore and take it forward

Discussion section should focus on the key findings related to the hypotheses along with the extant literature.

Implications should be reduced focusing on the critical consequences of the results.

Reviewer #3: I was unable to access the data provided. So I am assuming that the data were correctly computed and reported in the manuscript. Attached are my recommended edits for the authors line by line. In addition, all figures need to include footnotes re: level of statistical significance.

Reviewer #4: Thank you for the opportunity to review your article, “Promoting Teaching Innovation by Team Temporal Leadership: The Mediation of Job Autonomy and the modulation of Work Stress.” This paper seeks an association between team temporal leadership and teaching innovation through the mechanism of job autonomy. Further, it explores the moderating effect of work stress on that association. This excellent paper shows an interesting and novel association between these constructs. In particular, the identification of job autonomy as a mediating variable is an fascinating and insightful contribution to the literature.

General: I am not aware of the term “modulation” when referring to an interacting variable. I have always read and used the term “moderation.” This may be a language translation issue. The authors should ensure that the term is being used properly and consistently throughout the paper.

2.2 The team temporal leadership of school principals may affect teacher innovative work behavior: As it is the focus of the paper, I would have appreciated a more thorough explanation of team temporal leadership.

2.4 The moderation of work stress: There should be citations for the assertions in lines 342-345.

I believe Hypothesis 4 is needlessly complex and could be ended at the colon. The mechanisms of how work stress moderates the association of team temporal leadership and teaching innovation are clearly explained in the Discussion.

3. Methods: The primary thing I would like to see addressed in this paper is the distinction between administrators’ actual team temporal leadership vs. the participants’ perception of their team temporal leadership. I am concerned that teaching innovation is higher and work stress lower for teachers with higher self-efficacy, and those teachers may also be more likely to recognize and/or reap the benefits of team temporal leadership. Perhaps this could be controlled for if there are data on the participants’ sense of self-efficacy or other measures of teaching effectiveness.

3.1 Participants: I would like to know how many different schools are represented in the sample.

3.2.4 Work stress: In lines 416-417, the sample question from the work stress questionnaire is not clear.

5.1 Theoretical Implications: I am glad you made the assertion in lines 573-574 that “Temporal leadership is not synonymous with rigid time management that turns a blind eye to individual differences among team members.” Faced with the findings of this paper, disciples of Taylor’s Scientific Management might reflexively think that micro-management of staff’s schedule is an appropriate response. In fact, as you write, it is not.

Reviewer #5: Very interesting, well researched and well written study. Write-up is engaging to people interested in the different areas touched by this research such as leadership styles and facilitators for innovation take-up and implementation even if they are not working in the educational context. I have only included some minor editorial changes for the author's consideration.

Several interesting contributors were touched and discussed both at the literature review level as well during the discussion of the findings

The only major perspective that I missed seeing even a mention of is the effect of allowance for failure in innovation. Employees need to feel 'safe' that they will not be penalized should failure be the result of innovation. This perspective has also been researched extensively in its interaction with the take-up and implementation of innovation.

Reviewer #6: The study examines impact of temporal leadership on innovative teaching. This relationship is mediated via job autonomy for teaching which has been proven to improve teaching outcomes.

The following are my comments:

1. definition of temporal leadership can be more concise. Authors may cut down on extraneous details.

2. methods and results: the authors have done an excellent job in writing the study methods and explaining the results. It was easy to folllow, I do not have any objections or concerns.

3. discussion: this line is not clear "515 Our findings reveal a clear message: we may not be able to schedule teaching

516 innovation, but we can schedule for it." Please reframe.

Some typos:

line 47-- spacing between words

line 467--upper case "Job"

This is good work, can revise the above and resubmit. The manuscript is ready for publication in my opinion after addressing the above comments.

Reviewer #7: An important component of the sample should be described in more detail in the paper. Workplace autonomy and educational innovation have been shown to be stronger in schools where legislation on educational activities is more permissive. This is typically the case for private institutions, less so for public institutions. Therefore, it would be worthwhile to present in a few sentences that the sample includes responses from teachers in public or private schools. What is the proportion of responses received from teachers in private and public institutions? Is there a relevant difference between the answers received depending on whether the instructor giving the answer teaches in a public or private institution?

6. PLOS authors have the option to publish the peer review history of their article (what does this mean?). If published, this will include your full peer review and any attached files.

Reviewer #1: **Yes: **Ogechukwu Patrick (PhD)

Reviewer #2: No

Reviewer #3: No

Reviewer #4: No

Reviewer #5: No

Reviewer #6: No

Reviewer #7: No

---

## [Author Response · Author response to Decision Letter 0]

11 Jun 2022

Dear Editor(s) and reviewers:

 We thank you, sincerely, for handling our manuscript and giving us valuable suggestions. We are encouraged by your positive feedbacks, including the novelty of research question, comprehensiveness of literature review, and importance of research findings, as well as the rigidity of the methodology highlighted by most of the reviewers. 

 We valued the suggestions deeply, and made accordingly one-to-one response to each of the concerns. We have integrated our replies into our revised manuscript, with changes highlighted at corresponding locations. We hope our revised manuscript has been substantially improved, and look forward to your good news.

 Please see our reponses in our attached doc file in 'response to reviewers', which could also be seen at the end of the PDF file built along with the submission.

Thank you very much.

Guiqin, Zhu

---

## [Editor Report · Decision Letter 1]

27 Jun 2022

Promoting Teaching Innovation of Chinese Public-School Teachers by Team Temporal Leadership: The Mediation of Job Autonomy and the moderation of Work Stress

PONE-D-22-11897R1

Dear Dr. Guiqin Zhu,

We’re pleased to inform you that your manuscript has been judged scientifically suitable for publication and will be formally accepted for publication once it meets all outstanding technical requirements.

Kind regards,

Rogis Baker, Ph.D

Academic Editor

PLOS ONE
---

## [Editor Report · Acceptance letter]

30 Jun 2022

PONE-D-22-11897R1 

Promoting Teaching Innovation of Chinese Public-School Teachers by Team Temporal Leadership: The Mediation of Job Autonomy and the moderation of Work Stress 

Dear Dr. Zhu:

I'm pleased to inform you that your manuscript has been deemed suitable for publication in PLOS ONE. Congratulations! Your manuscript is now with our production department. 

Kind regards, 

on behalf of

Dr. Rogis Baker 

Academic Editor

PLOS ONE